# DENSITY RATIO ESTIMATION-BASED BAYESIAN OPTIMIZATION WITH SEMI-SUPERVISED LEARNING

## ABSTRACT

Bayesian optimization has attracted huge attention from diverse research areas in science and engineering, since it is capable of finding a global optimum of an expensive-to-evaluate black-box function efficiently. In general, a probabilistic regression model, e.g., Gaussian processes and Bayesian neural networks, is widely used as a surrogate function to model an explicit distribution over function evaluations given an input to estimate and a training dataset. Beyond the probabilistic regression-based Bayesian optimization, density ratio estimation-based Bayesian optimization has been suggested in order to estimate a density ratio of the groups relatively close and relatively far to a global optimum. Developing this line of research further, a supervised classifier can be employed to estimate a class probability for the two groups instead of a density ratio. However, the supervised classifiers used in this strategy are prone to be overconfident for a global solution candidate. To solve this problem, we propose density ratio estimation-based Bayesian optimization with semi-supervised learning. Finally, we demonstrate the experimental results of our methods and several baseline methods in two distinct scenarios with unlabeled point sampling and a fixed-size pool.

## 1 INTRODUCTION

Bayesian optimization (Brochu et al., 2010; Garnett, 2023) has attracted immense attention from various research areas such as hyperparameter optimization (Bergstra et al., 2011), battery lifetime optimization (Attia et al., 2020), and chemical reaction optimization (Shields et al., 2021), since it is capable of finding a global optimum of an expensive-to-evaluate black-box function in a sample-efficient manner. As studied in previous literature on Bayesian optimization (Snoek et al., 2012; Martinez-Cantin et al., 2018; Springenberg et al., 2016; Hutter et al., 2011), a probabilistic regression model, which can estimate a distribution of function evaluations over inputs, is widely used as a surrogate function; Gaussian process (GP) regression (Rasmussen & Williams, 2006) is a predominant choice for the surrogate function. An analogy between probabilistic regression models in Bayesian optimization is that they rely on an explicit function over function evaluations $p(y \mid \mathbf{x}, \mathcal{D})$ given an input we desire to estimate, denoted as $\mathbf{x}$, and a training dataset $\mathcal{D}$.

Beyond the probabilistic regression-based Bayesian optimization, density ratio estimation (DRE)-based Bayesian optimization has been studied recently (Bergstra et al., 2011; Tiao et al., 2021). Furthermore, likelihood-free Bayesian optimization, which is equivalent to DRE-based Bayesian optimization with a particular utility function, has been proposed by Song et al. (2022). Bergstra et al. (2011) attempt to model two densities $p(\mathbf{x} \mid y \leq y^\dagger, \mathcal{D})$ and $p(\mathbf{x} \mid y > y^\dagger, \mathcal{D})$, where $y^\dagger$ is a threshold for dividing inputs to two groups that are relatively close and relatively far to a global solution, in order to estimate $\zeta$-relative density ratio (Yamada et al., 2011). On the other hand, instead of modeling two densities separately, Tiao et al. (2021); Song et al. (2022) estimate a density ratio using class-probability estimation (Qin, 1998). As discussed in the previous work, this line of research provides a new understanding of Bayesian optimization, which allows us to solve Bayesian optimization using binary classification. Moreover, it can reduce the amount of computations required for building surrogate functions.

However, the supervised classifier utilized in the DRE-based Bayesian optimization are prone to be overconfident for a global solution candidate (or a potential region that exists a global solution). In this paper, to solve this overconfidence problem, we propose a novel DRE-based

method with semi-supervised learning, named DRE-BO-SSL. Although our direct competitors, i.e., BORE (Tiao et al., 2021) and LFBO (Song et al., 2022), show their strengths through theoretical and empirical analyses, our algorithm betters an ability to consider a wider region that satisfies $p(\mathbf{x} \mid y \leq y^{\dagger}, \mathcal{D}) \geq p(\mathbf{x} \mid y > y^{\dagger}, \mathcal{D})$, than the competitors, as shown in Figure 1. By this intuitive example in Figure 1, we presume that DRE-BO-SSL appropriately balances exploration and exploitation rather than the existing methods. Compared to a supervised classifier, e.g., random forests (Breiman, 2001), gradient boosting (Friedman, 2001), and multi-layer perceptrons, our semi-supervised classifiers, i.e., label propagation (Zhu & Ghahramani, 2002) and label spreading (Zhou et al., 2003), are less confident in terms of the regions of global solution candidates by using unlabeled data points; see Figures 1 and 8 and Section 3 for detailed examples and analyses.

To make use of semi-supervised classifiers, we take into account two distinct scenarios with unlabeled point sampling and with a predefined fixed-size pool. For the first scenario, we randomly sample unlabeled data points from the truncated multivariate normal distribution using a minimax tilting method (Botev, 2017), in order that it is possible that a cluster assumption (Seeger, 2000) is assumed. Finally, we demonstrate that our method shows superior performance compared to the exiting methods in diverse experiments including synthetic benchmarks, Tabular Benchmarks (Klein & Hutter, 2019), NATS-Bench (Dong et al., 2021), and minimum multi-digit MNIST search.

To sum up, our contributions are itemized as follows: (i) we identify the overconfidence problem of supervised classifiers in DRE-based Bayesian optimization; (ii) we propose DRE-based Bayesian optimization with semi-supervised learning, named DRE-BO-SSL for two distinct scenarios with unlabeled point sampling and a predefined fixed-size pool; (iii) we demonstrate the effectiveness of our method in various experiments including NATS-Bench and minimum multi-digit MNIST search.

### 1.1 OVERCONFIDENCE PROBLEM

As illustrated in Figures 1 and 8, the supervised classifiers used in DRE-based Bayesian optimization suffer from the overconfidence problem. Interestingly, many deep learning models also share a similar overconfidence problem (Guo et al., 2017; Müller et al., 2019), due to various reasons, but primarily due to overparameterization.

However, we note that the definition of the overconfidence problem in DRE-based Bayesian optimization is slightly different from the problem in general classification. The definition in DRE-based Bayesian optimization does not imply that a single data point has a high probability for a particular class, but it implies that a few data points have high probabilities for a class of interest. More precisely, our definition indicates a problem of overconfidence over global solution candidates. For example, the definition in general classification includes a case that 50% of data points are biased to one class and the remainder is biased to another class. On the contrary, our definition does not include such a case and it only includes a case that a small number of data points (or a small region of a search space) are biased to some class, i.e., Class 1 in our problem.

By the aforementioned definition, at early iterations of Bayesian optimization, a supervised classifier tends to overfit to a small size of $\mathcal{D}_t$ due to a relatively large model capacity. This consequence makes a Bayesian optimization algorithm highly focus on exploitation. Similar to our observation, the imbalance of exploration and exploitation in the DRE-based approaches is also discussed in the recent work (Pan et al., 2023). Moreover, the consequence mentioned above is also different from the characteristics of probabilistic regression-based Bayesian optimization because the regression-based Bayesian optimization is capable of exploring unseen regions by dealing with uncertainties. Besides, even though a threshold $\zeta$ might be able to mitigate this problem, an overconfident supervised classifier is likely to keep getting stuck in a local optimum as $y^{\dagger}$ cannot change dramatically.

*We promise to release our implementation upon publication.*

## 2 BACKGROUND AND RELATED WORK

**Bayesian Optimization.** It is a principled and efficient approach to finding a global solution of a challenging objective, e.g., expensive-to-evaluate black-box functions (Brochu et al., 2010; Garnett, 2023). To focus on a probabilistic regression model as a surrogate function, we omit the details of

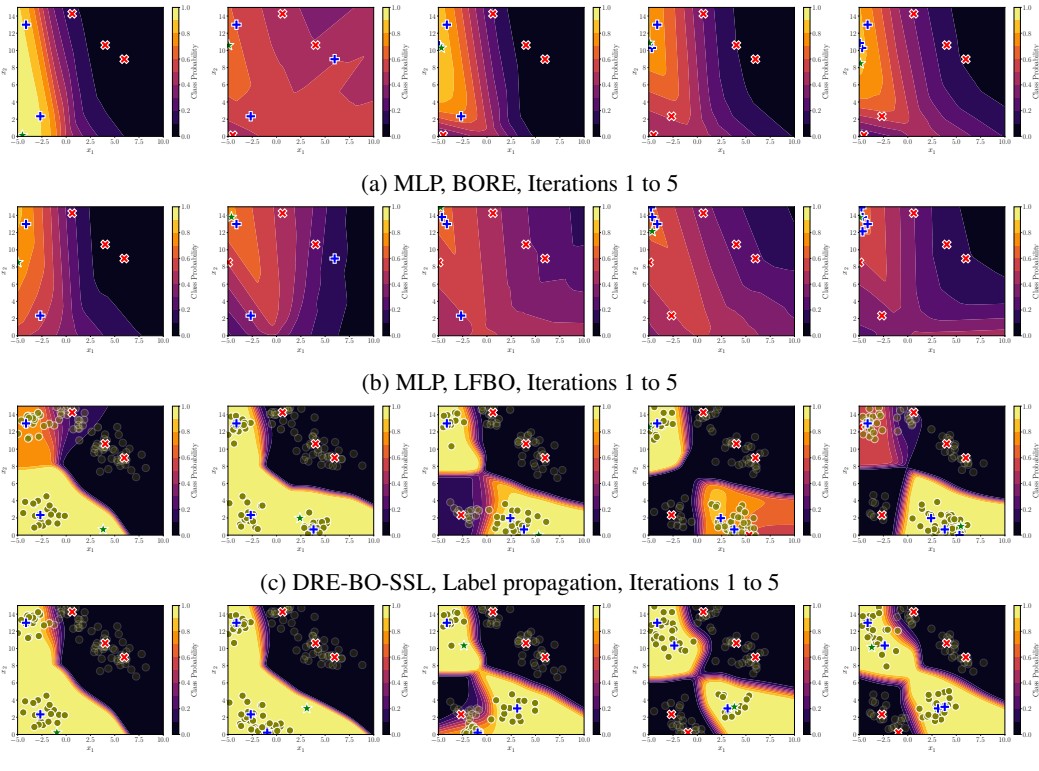

Figure 1: Comparisons of BORE and LFBO by multi-layer perceptrons, and DRE-BO-SSL with label propagation and label spreading for the Branin function. Each row shows Iterations 1 to 5 with five initial points. + (blue), x (red), and ⋆ (green) indicate data points of $y \leq y^{\dagger}$, data points of $y > y^{\dagger}$, and query points, respectively. Moreover, ○ (olive) stands for unlabeled points and its transparency represents the class probability predicted by a semi-supervised classifier. A query point is chosen by maximizing the output of the classifier. More results are shown in Figure 8.

Bayesian optimization here; see (Brochu et al., 2010; Garnett, 2023) for its details. In the Bayesian optimization community, GP regression (Rasmussen & Williams, 2006) is widely used as a surrogate function (Srinivas et al., 2010; Snoek et al., 2012) because of its flexibility with minimum assumptions on model and smoothness. While GP is a probable choice, Bayesian optimization with diverse surrogate functions such as Student-$t$ process regression (Martinez-Cantin et al., 2018), Bayesian neural networks (Springenberg et al., 2016), and tree-based models (Hutter et al., 2011; Kim & Choi, 2022) has been proposed. An analogy between such models is that they model $p(y \mid \mathbf{x}, \mathcal{D})$ explicitly, so that it can be used to define an acquisition function with the statistics of $p(y \mid \mathbf{x}, \mathcal{D})$.

**Density-Ratio Estimation.** Whereas knowing a data distribution $p(\mathbf{x})$ is important, it is difficult to directly estimate $p(\mathbf{x})$ (Sugiyama et al., 2012). For specific machine learning problems such as importance sampling (Kloek & van Dijk, 1978) and mutual information estimation (Bishop, 2006), we can bypass direct density estimation and estimate a density ratio. In Bayesian optimization, Bergstra et al. (2011) propose a strategy with tree-structured Parzen estimators to estimate a density ratio as an alternative to probabilistic regression-based acquisition functions. In addition, the work (Tiao et al., 2021; Song et al., 2022) suggests methods with binary classifiers in order to estimate class probabilities as a density ratio; see Section 3 for their details.

**Semi-Supervised Learning.** It is a learning scheme with both labeled and unlabeled data (Zhu, 2005; Chapelle et al., 2006; Bengio et al., 2006). To cooperate with labeled and unlabeled data, this strategy generally utilizes geometry of data points or connectivity between points, and assigns a pseudo-label to unlabeled data, which referred to as transductive learning (Gammerman et al., 1998). As a semi-supervised learning method on a similarity graph, Zhu & Ghahramani (2002)

propose a label propagation algorithm which iteratively propagates the labels of unlabeled data points using labeled data. Zhou et al. (2003) compute the labels of labeled and unlabeled data points by a weighted iterative algorithm with initial labels. Belkin & Niyogi (2002) predict pseudo-labels by finding a linear combination of a few smallest eigenvectors of the graph Laplacian.

## 3 DRE-BASED BAYESIAN OPTIMIZATION

Unlike probabilistic regression-based Bayesian optimization, DRE-based Bayesian optimization employs a density ratio-based acquisition function, defined with a density $p(\mathbf{x} \mid y \leq y^\dagger, \mathcal{D}_t)$, where $\mathbf{x}$ is a $d$-dimensional vector, $y$ is its function evaluation, $y^\dagger$ is a threshold, and $\mathcal{D}_t = \{(\mathbf{x}_i, y_i)\}_{i=0}^t$ is a dataset of $t + 1$ pairs of data point and evaluation. In particular, the work by Bergstra et al. (2011) defines an acquisition function based on $\zeta$-relative density ratio (Yamada et al., 2011):

$$A(\mathbf{x} \mid \zeta, \mathcal{D}_t) = \frac{p(\mathbf{x} \mid y \leq y^\dagger, \mathcal{D}_t)}{\zeta p(\mathbf{x} \mid y \leq y^\dagger, \mathcal{D}_t) + (1 - \zeta)p(\mathbf{x} \mid y > y^\dagger, \mathcal{D}_t)}, \tag{1}$$

where $\zeta = p(y \leq y^\dagger) \in [0, 1)$ is a threshold ratio. We need to find a maximizer of (1), by optimizing the following composite function: $h(p(\mathbf{x} \mid y \leq y^\dagger, \mathcal{D}_t)/p(\mathbf{x} \mid y > y^\dagger, \mathcal{D}_t))$, where $h(x) = (\zeta + (1 - \zeta)x^{-1})^{-1}$. Since $h$ is a strictly increasing function, we can directly maximize $p(\mathbf{x} \mid y \leq y^\dagger, \mathcal{D}_t)/p(\mathbf{x} \mid y > y^\dagger, \mathcal{D}_t)$. In the previous work (Bergstra et al., 2011), two tree-structured Parzen estimators are used to estimate the respective densities, $p(\mathbf{x} \mid y \leq y^\dagger, \mathcal{D}_t)$ and $p(\mathbf{x} \mid y > y^\dagger, \mathcal{D}_t)$.

While the work (Bergstra et al., 2011) utilizes two distinct tree-structured Parzen estimators, Tiao et al. (2021) propose a method to directly estimate (1) using class-probability estimation (Qin, 1998; Sugiyama et al., 2012), which is called BORE. Since it can be formulated as a problem of binary classification in which Class 1 is a group of the top $\zeta$ of $\mathcal{D}_t$ and Class 0 is a group of the bottom $1 - \zeta$ of $\mathcal{D}_t$ in terms of function evaluations, the acquisition function defined in (1) induces the following:

$$A(\mathbf{x} \mid \zeta, \mathcal{D}_t) = \frac{p(\mathbf{x} \mid z = 1)}{\zeta p(\mathbf{x} \mid z = 1) + (1 - \zeta)p(\mathbf{x} \mid z = 0)}. \tag{2}$$

By the Bayes' theorem, (2) becomes $A(\mathbf{x} \mid \zeta, \mathcal{D}_t) = \zeta^{-1}p(z = 1 \mid \mathbf{x})/(p(z = 1 \mid \mathbf{x}) + p(z = 0 \mid \mathbf{x}))$. Therefore, a class probability over $\mathbf{x}$ for Class 1 is considered as an acquisition function; it is simply denoted as $A(\mathbf{x} \mid \zeta, \mathcal{D}_t) = \zeta^{-1}\pi(\mathbf{x})$. Eventually, the class probability is estimated by various off-the-shelf classifiers such as random forests and multi-layer perceptrons.

Song et al. (2022) suggest a general framework of likelihood-free Bayesian optimization, called LFBO:

$$A(\mathbf{x} \mid \zeta, \mathcal{D}_t) = \arg\max_{S: \mathcal{X} \to \mathbb{R}} \mathbb{E}_{\mathcal{D}_t}[u(y; y^\dagger)f'(S(\mathbf{x})) - f^*(f'(S(\mathbf{x})))], \tag{3}$$

which is versatile for any non-negative utility function $u(y; y^\dagger)$, where $f$ is a strictly convex function and $f^*$ is the convex conjugate of $f$. By the properties of LFBO, it is equivalent to an expected utility-based acquisition function. Along with the general framework, Song et al. (2022) prove that BORE is equivalent to the probability of improvement (Kushner, 1964) and LFBO with $u(y; y^\dagger) = \mathbb{I}(y \leq y^\dagger)$ is also equivalent to the probability of improvement, where $\mathbb{I}$ is an indicator function. Moreover, they show that their method with $u(y; y^\dagger) = \max(y^\dagger - y, 0)$, which can be implemented as weighted binary classification, is equivalent to the expected improvement (Jones et al., 1998).

## 4 DRE-BASED BAYESIAN OPTIMIZATION WITH SEMI-SUPERVISED LEARNING

We introduce DRE-based Bayesian optimization with semi-supervised learning, named DRE-BO-SSL, by following the previous studies in DRE-based and likelihood-free Bayesian optimization (Bergstra et al., 2011; Tiao et al., 2021; Song et al., 2022), which were discussed in Section 3.

Similar to the standard Bayesian optimization and existing DRE-based Bayesian optimization, DRE-BO-SSL iterates the undermentioned steps as presented in Algorithm 1. Firstly, a threshold $y_t^\dagger$ is calculated by considering $y_{1:t}$ with $\zeta$. Secondly, labels $\mathbf{C}_t$ of data points in $\mathcal{D}_t$ are assigned to one of two classes; a group of $y \leq y_t^\dagger$ is assigned to Class 1 and a group of $y > y_t^\dagger$ is assigned to Class 0.

---

**Algorithm 1** DRE-BO-SSL

---

**Input:** Iteration budget $T$, a search space $\mathcal{X}$, a black-box objective $f$, a threshold ratio $\zeta$, a semi-supervised classifier $\pi_{\mathbf{C}}$, and unlabeled data points $\mathbf{X}_u$ if available and the number of unlabeled data points $n_u$, otherwise.
**Output:** Best candidate $\mathbf{x}^+$.
 1: Initialize $\mathcal{D}_0 = \{(\mathbf{x}_0, y_0)\}$ by randomly selecting $\mathbf{x}_0$ from $\mathcal{X}$ and evaluating $\mathbf{x}_0$ by $f$.
 2: **for** $t = 0$ **to** $T - 1$ **do**
 3:     Calculate a threshold $y_t^\dagger$ using $\zeta$.
 4:     Assign labels $\mathbf{C}_t$ of data points in $\mathcal{D}_t$ with $y_t^\dagger$.
 5:     **if** $\mathbf{X}_u$ are not available **then**
 6:         Sample $n_u$ unlabeled data points $\mathbf{X}_u$ from $\mathcal{X}$.
 7:     **end if**
 8:     Estimate pseudo-labels $\widehat{\mathbf{C}}_t$ by following the procedure in Algorithm 2.
 9:     Choose the next query point $\mathbf{x}_{t+1}$ by maximizing $\pi_{\widehat{\mathbf{C}}_t}(\mathbf{x}; \zeta, \mathcal{D}_t, \mathbf{X}_u)$ for $\mathbf{x} \in \mathcal{X}$.
10:     Evaluate $\mathbf{x}_{t+1}$ by $f$ and update $\mathcal{D}_{t+1}$.
11: **end for**
12: Determine the best candidate $\mathbf{x}^+$, considering $y_{0:T}$.

---

If we are given unlabeled data points $\mathbf{X}_u$, the corresponding points $\mathbf{X}_u$ are used, but if not available it samples $n_u$ unlabeled data points $\mathbf{X}_u$ from $\mathcal{X}$ by utilizing a strategy described in Section 4.2. Then, it estimates pseudo-labels $\widehat{\mathbf{C}}_t$ of a semi-supervised learning model by following the procedure in Section 4.1 and Algorithm 2. Using $\widehat{\mathbf{C}}_t$, it chooses the next query point $\mathbf{x}_{t+1}$:

$$\mathbf{x}_{t+1} = \arg\max_{\mathbf{x} \in \mathcal{X}} \pi_{\widehat{\mathbf{C}}_t}(\mathbf{x}; \zeta, \mathcal{D}_t, \mathbf{X}_u), \tag{4}$$

where $\pi_{\widehat{\mathbf{C}}_t}(\mathbf{x}; \zeta, \mathcal{D}_t, \mathbf{X}_u)$ predicts a class probability over $\mathbf{x}$ for Class 1; see (9).

We adopt a multi-started local optimization technique, e.g., L-BFGS-B (Byrd et al., 1995), to solve (4). However, a flat landscape of $\pi_{\widehat{\mathbf{C}}_t}(\mathbf{x}; \zeta, \mathcal{D}_t, \mathbf{X}_u)$ over $\mathbf{x}$ may occur, so that optimization performance can be degraded. To tackle this issue, a simple heuristic of randomly selecting a query point among points with identical highest class probabilities complements our method. Since the multi-started technique is utilized and the output of $\pi_{\widehat{\mathbf{C}}_t}$ is bounded in $[0, 1]$, a flat landscape is easily recognized by comparing the outcomes of the multi-started strategy.

## 4.1 Label Propagation and Label Spreading

Here we describe semi-supervised learning techniques (Zhu, 2005; Chapelle et al., 2006; Bengio et al., 2006) to build DRE-BO-SSL. We cover a transductive setting (Gammerman et al., 1998), which is to label unlabeled data by utilizing given labeled data, and then an inductive setting, which is to predict any point using pseudo-labels of unlabeled and labeled data.

Suppose that each data point is defined on a $d$-dimensional compact space $\mathcal{X} \subset \mathbb{R}^d$. We consider $n_l$ labeled points $\mathbf{X}_l \in \mathbb{R}^{n_l \times d}$, their corresponding labels $\mathbf{C}_l \in \mathbb{R}^{n_l \times c}$, and $n_u$ unlabeled points $\mathbf{X}_u \in \mathbb{R}^{n_u \times d}$, where $c$ is the number of classes. $\mathbf{X}_l$ and $\mathbf{C}_l$ are query points that have already been evaluated and their class labels; we define $\mathbf{X}_l = [\mathbf{x}_0, \ldots, \mathbf{x}_{n_l-1}]^\top$ for $\mathcal{D}_t = \{(\mathbf{x}_i, y_i)\}_{i=0}^{n_l-1}$. For the sake of brevity, the concatenated data points of $\mathbf{X}_l$ and $\mathbf{X}_u$ are defined as $\mathbf{X} = [\mathbf{X}_l; \mathbf{X}_u] \in \mathbb{R}^{(n_l+n_u) \times d}$. Note that in our problem $c = 2$ because we address the problem with two classes.

As shown in Algorithm 2, we first initialize labels to propagate $\widehat{\mathbf{C}} \in \mathbb{R}^{(n_l+n_u) \times 2}$; it is initialized as $\widehat{\mathbf{C}} = [\mathbf{c}_0, \ldots, \mathbf{c}_{n_l-1}, \mathbf{c}_{n_l}, \ldots, \mathbf{c}_{n_l+n_u-1}]^\top$, where $\mathbf{c}_0, \ldots, \mathbf{c}_{n_l-1}$ are one-hot representations $[\mathbf{C}_l]_{1:}, \ldots, [\mathbf{C}_l]_{n_l:}$; and $\mathbf{c}_{n_l}, \ldots, \mathbf{c}_{n_l+n_u-1}$ are zero vectors. Denote that $[\mathbf{C}]_{i:}$ is $i$th row of $\mathbf{C}$. Then, we compute a similarity between two data points $\mathbf{x}_i, \mathbf{x}_j \in \mathcal{X}$, so that we compare all pairs in $\mathbf{X}$. For example, one of popular similarity functions, i.e., a radial basis function kernel, can be used:

$$w_{ij} = \exp\left(-\beta \|\mathbf{x}_i - \mathbf{x}_j\|_2^2\right), \tag{5}$$

where $\beta$ is a free parameter given. As discussed in the work (Zhu & Ghahramani, 2002), we can learn $\beta$ in (5) by minimizing an entropy for propagated labels $\widehat{\mathbf{C}}$:

$$H(\widehat{\mathbf{C}}) = -\sum_{i=1}^{n_l+n_u} [\widehat{\mathbf{C}}]_{i:}^\top \log[\widehat{\mathbf{C}}]_{i:}. \tag{6}$$

See Figure 4 for the results of learning $\beta$ and Section E for analysis on the effects of $\beta$. By (5), we compute a transition probability from $\mathbf{x}_j$ to $\mathbf{x}_i$ by $p_{ij} = w_{ij}/\sum_{k=1}^{n_l+n_u} w_{kj}$. Note that similarities $\mathbf{W} \in \mathbb{R}^{(n_l+n_u)\times(n_l+n_u)}$ and transition probabilities $\mathbf{P} \in \mathbb{R}^{(n_l+n_u)\times(n_l+n_u)}$ are defined, where $[\mathbf{W}]_{ij} = w_{ij}$ and $[\mathbf{P}]_{ij} = p_{ij}$. Moreover, by the definition of $p_{ij}$, $\mathbf{P} = \mathbf{D}^{-1}\mathbf{W}$, where $\mathbf{D}$ is a degree matrix whose diagonal entry is defined as $[\mathbf{D}]_{ii} = \sum_{j=1}^{n_l+n_u}[\mathbf{W}]_{ij}$.

With initial $\widehat{\mathbf{C}}$ and $\mathbf{P}$, we repeat the following steps: (i) computing the next $\widehat{\mathbf{C}}$; (ii) normalizing $\widehat{\mathbf{C}}$ row-wise, until a change of $\widehat{\mathbf{C}}$ converges to a tolerance $\varepsilon$ or the number of iterations propagated reaches to maximum iterations $\tau$. In particular, label propagation (Zhu & Ghahramani, 2002) updates $\widehat{\mathbf{C}}$ and constrains the labels of labeled data to $\mathbf{C}_l$:

$$\widehat{\mathbf{C}}_{t+1} \leftarrow \mathbf{P}^\top \widehat{\mathbf{C}}_t \quad \text{and} \quad [\widehat{\mathbf{C}}_{t+1}]_{i:} \leftarrow [\mathbf{C}_l]_{i:}, \tag{7}$$

for $i \in [n_l]$ at Iteration $t$. On the other hand, label spreading (Zhou et al., 2003) propagates $\widehat{\mathbf{C}}$ by allowing a change of the labels of labeled data with a clamping factor $\alpha \in (0,1)$:

$$\widehat{\mathbf{C}}_{t+1} \leftarrow \alpha \mathbf{D}^{-1/2}\mathbf{W}\mathbf{D}^{-1/2}\widehat{\mathbf{C}}_t + (1-\alpha)\widehat{\mathbf{C}}_0, \tag{8}$$

where $\widehat{\mathbf{C}}_0$ is the initial propagated labels defined Line 1 of Algorithm 2. Note that $\mathbf{D}^{-1/2}\mathbf{W}\mathbf{D}^{-1/2}$ can be pre-computed before the loop defined from Line 4 to Line 7 of Algorithm 2.

By the nature of transductive setting, it only predicts the labels of particular data using the known data (Gammerman et al., 1998), which implies that it cannot estimate a categorical distribution of unseen data. To enable it, given unseen $\mathbf{x}$, we define an inductive model with $\widehat{\mathbf{C}}$:

$$\widehat{c}_i = \frac{\mathbf{w}^\top [\widehat{\mathbf{C}}]_{:i}}{\sum_{j=1}^c \mathbf{w}^\top [\widehat{\mathbf{C}}]_{:j}}, \tag{9}$$

for $i \in [2]$, where $\mathbf{w} \in \mathbb{R}^{n_l+n_u}$ is similarities of $\mathbf{x}$ and $\mathbf{X}$ by (5). $[\widehat{\mathbf{C}}]_{:i}$ denotes $i$th column of $\widehat{\mathbf{C}}$. This inductive model is better than or equivalent to other classifiers without unlabeled data in certain circumstances; its analysis is provided in Section C.

## 4.2 UNLABELED POINT SAMPLING

As described above, if unlabeled points are not available, we need to generate unlabeled points under a transductive learning scheme. However, it is a challenging problem unless we know a landscape of $\pi_{\mathbf{C}}$ adequately. Many studies (Seeger, 2000; Rigollet, 2007; Singh et al., 2008; Ben-David et al., 2008; Carmon et al., 2019; Wei et al., 2020; Zhang et al., 2022) investigate how unlabeled data can affect a model and whether unlabeled data helps improve the model or not.

In order to make use of the theoretical findings of previous literature, we define a cluster assumption:

**Assumption 4.1** (Cluster assumption in (Seeger, 2000)). *Two points $\mathbf{x}_i, \mathbf{x}_j \in \mathcal{X}$ should belong to the same label if there is a path between $\mathbf{x}_i$ and $\mathbf{x}_j$ which passes only through regions of relatively high $P(X)$, where $P$ is a distribution over a random variable $X \in \mathcal{X}$.*

By Assumption 4.1, the idea of clustering on the Euclidean space or spectral clustering on a graph can be naturally applied in semi-supervised learning (Seeger, 2000; Joachims, 2003), which is not the scope of this work.

To build DRE-BO-SSL associated with Assumption 4.1, we sample unlabeled data points from the truncated multivariate normal distribution so that each sample is in a compact $\mathcal{X}$:

$$f(\mathbf{z}) = \frac{\exp(-\frac{1}{2}\mathbf{z}^\top \mathbf{z})\mathbb{I}(\mathbf{l} \leq \mathbf{A}\mathbf{z} \leq \mathbf{u})}{P(\mathbf{l} \leq \mathbf{A}Z \leq \mathbf{u})}, \tag{10}$$

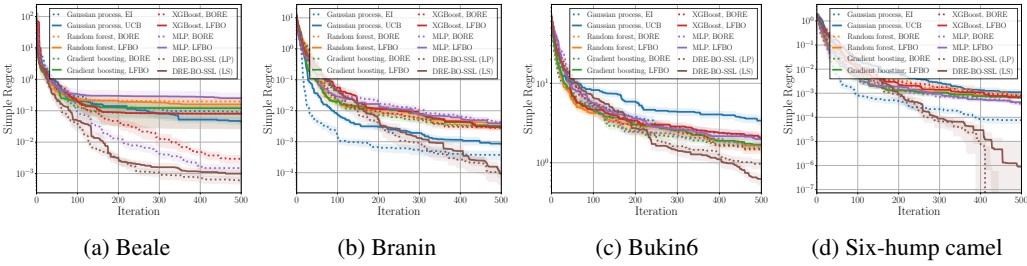

Figure 2: Results with 20 repeated experiments on synthetic benchmark functions for a scenario with unlabeled point sampling. LP and LS stand for label propagation and label spreading.

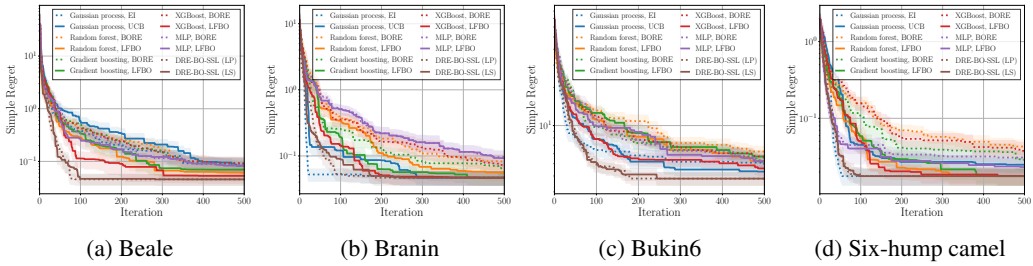

Figure 3: Results with 20 repeated experiments on synthetic benchmark functions for a scenario with fixed-size pools. LP and LS stand for label propagation and label spreading.

where $\mathbf{l}, \mathbf{u} \in \mathbb{R}^d$ are lower and upper bounds, $\boldsymbol{\Sigma} = \mathbf{A}\mathbf{A}^\top$ is a covariance matrix, $\mathbb{I}$ is an indicator function, and $Z \sim \mathcal{N}(\mathbf{0}, \mathbf{I}_d)$ is a random variable. It is challenging to calculate a denominator of (10), $P(\mathbf{l} \leq \mathbf{A}^\top Z \leq \mathbf{u})$, and simulate from $f(\mathbf{z})$ because an integration of the denominator and an accept-reject sampling strategy from $f(\mathbf{z})$ are cumbersome in this multi-dimensional case. To effectively sample from the truncated multivariate normal distribution, we adopt a minimax tilting method (Botev, 2017). Compared to a method by Genz (1992), it yields a high acceptance rate and accurate sampling. In this paper $\boldsymbol{\Sigma}$ is set as an identity matrix, and $\mathbf{l}$ and $\mathbf{u}$ are determined by a search space. We will provide more detailed discussion on point sampling in Sections 6 and F.

## 5 EXPERIMENTS

We test DRE-BO-SSL and baseline methods in the following optimization problems: synthetic benchmarks for a scenario with unlabeled point sampling, and synthetic benchmarks, Tabular Benchmarks (Klein & Hutter, 2019), NATS-Bench (Dong et al., 2021), and minimum multi-digit MNIST search for a scenario with a fixed-size pool. Note that Tabular Benchmarks, NATS-Bench, and minimum multi-digit MNIST search are defined with a fixed number of possible solution candidates, which implies that they are considered as combinatorial optimization problems. By following the previous work (Tiao et al., 2021; Song et al., 2022), we set a threshold ratio as $\zeta = 0.33$ for all experiments. To solve (4), we use L-BFGS-B (Byrd et al., 1995) with 1000 different initializations. All experiments are repeated 20 times with 20 fixed random seeds and the sample mean and the standard error of the sample mean are reported. Other missing details including the details of the competitors of our methods are presented in Section D.

### 5.1 A SCENARIO WITH UNLABELED POINT SAMPLING

**Synthetic Benchmarks.** We run several synthetic functions for our methods and the baseline methods. For unlabeled point sampling, we sample 100 points from $\mathcal{N}(\mathbf{x}_1, \mathbf{I}_d), \dots, \mathcal{N}(\mathbf{x}_{n_l}, \mathbf{I}_d)$, where $\lfloor n_u/n_l \rfloor$ or $\lfloor n_u/n_l \rfloor + 1$ points are sampled from each truncated distribution, so that $n_u$ points are sampled in total. As shown in Figure 2, our methods outperform the baseline methods incorporating labeled data with unlabeled points. Interestingly, out methods beat GP-based Bayesian optimization. It implies that ours can fairly balance exploration and exploitation. Furthermore, we present the results of learning $\beta$ in Figure 4, where $\beta$ is adaptively selected by (6) every iteration.

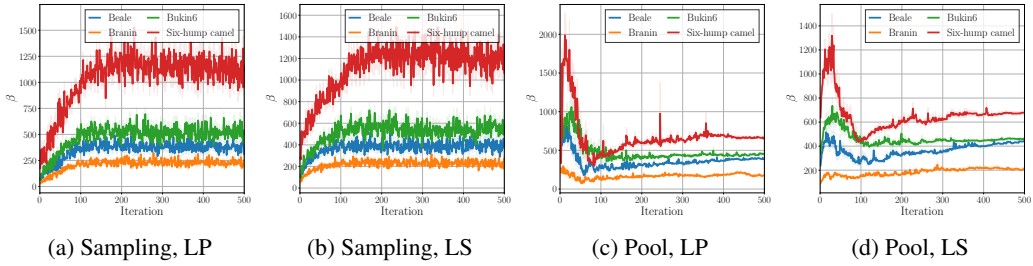

|  |  |  |  |
|---|---|---|---|
| (a) Sampling, LP | (b) Sampling, LS | (c) Pool, LP | (d) Pool, LS |

Figure 4: Results with 20 repeated experiments on learning $\beta$ for label propagation (LP) and label spreading (LS). Sampling and pool indicate the experiments in Sections 5.1 and 5.2, respectively.

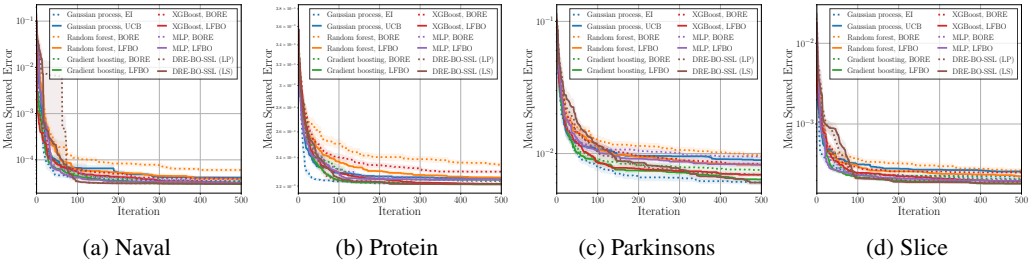

|  |  |  |  |
|---|---|---|---|
| (a) Naval | (b) Protein | (c) Parkinsons | (d) Slice |

Figure 5: Results with 20 repeated experiments on Tabular Benchmarks for a scenario with fixed-size pools. LP and LS stand for label propagation and label spreading.

## 5.2 SCENARIOS WITH FIXED-SIZE POOLS

**Synthetic Benchmarks.** Several synthetic benchmark functions are tested for our methods and the baseline methods. To generate a fixed-size pool for each benchmark, we uniformly sample 1000 points from a bounded search space before an optimization round is started. As presented in Figure 3, our methods perform better than the baseline methods. It implies that the use of unlabeled data helps improve optimization performance. Also, the results of learning $\beta$ are demonstrated in Figure 4. Learned $\beta$ is likely to converge to some value as iterations proceed according to the empirical results.

**Tabular Benchmarks.** Comparisons of our methods and the existing methods are carried out in these hyperparameter optimization benchmarks (Klein & Hutter, 2019), as reported in Figure 5. We can benchmark a variety of machine learning models, which are defined with specific hyperparameters and trained on one of four datasets: naval propulsion, protein structure, parkinsons telemonitoring, and slice localization. There exist 62,208 models, which are used as a pool in this paper, for each dataset. Our algorithms show superior performance compared to other approaches. Similar to the synthetic functions, we argue that the adoption of a predefined pool leverages its performance. In some cases, GP-based Bayesian optimization is better than our methods.

**NATS-Bench.** NATS-Bench (Dong et al., 2021), which is the up-to-date version of NAS-Bench-201 (Dong & Yang, 2019), is used to test our methods and the baseline methods. NATS-Bench is a neural architecture search benchmark with three popular datasets: CIFAR-10, CIFAR-100, and ImageNet-16-120, and it has 32,768 architectures, i.e., a fixed-size pool in this paper, for each dataset. Similar to the experiments mentioned above, our methods work well in three datasets, compared to the existing methods; see Figure 6 for the results.

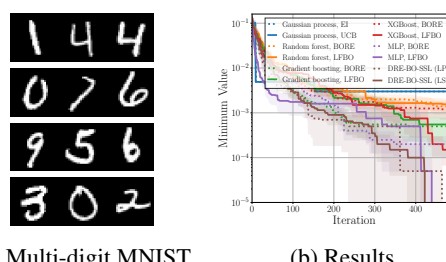

| (a) Multi-digit MNIST | (b) Results |
|---|---|

Figure 7: Minimum multi-digit MNIST search. Figure 7a shows some examples of multi-digit MNIST and Figure 7b demonstrates the results with 20 repeated experiments.

**Minimum Multi-Digit MNIST Search.** This task, which is proposed in this work, is to find a minimum multi-digit number, where a fixed number of multi-digit MNIST images are given. As visualized in Figure 7, three random images in the

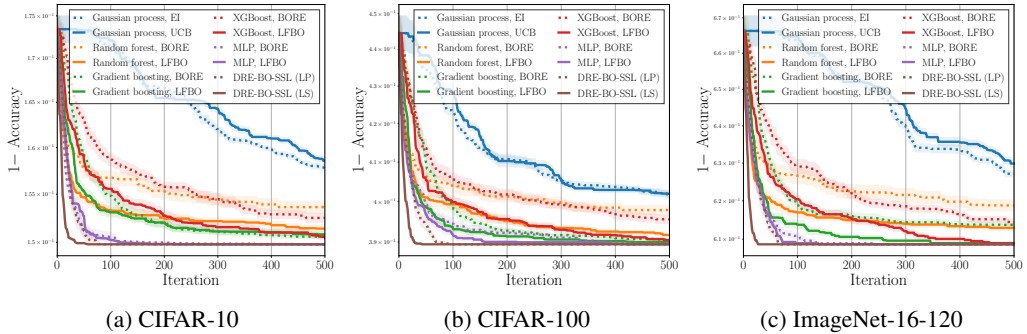

|  |  |  |
|:---:|:---:|:---:|
| (a) CIFAR-10 | (b) CIFAR-100 | (c) ImageNet-16-120 |

Figure 6: Results of NATS-Bench for a scenario with fixed-size pools. We repeat all experiments 20 times. LP and LS stand for label propagation and label spreading.

MNIST dataset (LeCun et al., 1998) are concatenated. Eventually, "000" and "999" are global minimum and global maximum, respectively. Since inputs are images and each concatenated image consists of three different digit images, this optimization problem is challenging. The size of a fixed-size pool is 80,000. As shown in Figure 7, our methods show satisfactory performance compared to other baselines.

## 6  DISCUSSION

**Flat Landscape of Class Probabilities over Inputs.**   Regardless of the use of either supervised or semi-supervised classifier, a flat landscape of class probabilities can occur in the framework of DRE-based Bayesian optimization. To overcome the issue of optimizing a flat landscape, we add a simple heuristic of selecting a random query point from points with identical highest class probabilities if the landscape is flat, as described in Section 4. Since off-the-shelf local optimization methods struggle to optimize a flat landscape, this simple heuristic is effective.

**Effects of the Number of Points and Sampling Distributions for Unlabeled Points.**   We choose a distribution for unlabeled point sampling as the truncated multivariate normal distribution in order to satisfy the cluster assumption. To analyze our algorithm DRE-BO-SSL thoroughly, we demonstrate the effects of the number of sampled points and sampling distributions for unlabeled points in Section F, varying the number of unlabeled points and utilizing diverse sampling distributions, i.e., uniform distributions, Halton sequences (Halton, 1960), and Sobol' sequences (Sobol', 1967).

**Effects of Pool Sampling.**   Because the computational complexity of label propagation and label spreading depends on a pool size, we need to reduce a pool size appropriately in order to speed up the algorithms. Therefore, we uniformly sample a subset of the fixed-size set, which is used as unlabeled points. Analysis on the effects of pool sampling is demonstrated in Section G.

**General Framework of DRE-BO-SSL.**   As a future research direction, we expect that a general framework of DRE-BO-SSL can be defined, similar to a likelihood-free approach by Song et al. (2022). However, it is difficult to define an utility function involved in both labeled and unlabeled data. For example, if we assume that an utility is $u(y; y^{\dagger}) = \max(y^{\dagger} - y, 0)$, $y$ for an unlabeled data point is unknown. Although it depends on the form of utility function, we need to define $y$ of an unlabeled data point by utilizing the information we have if the utility function is related to $y$.

## 7  CONCLUSION

In this paper we have proposed a DRE-based Bayesian optimization framework with semi-supervised learning, named DRE-BO-SSL. Unlike the existing work such as BORE and LFBO, our methods make use of a semi-supervised classifier, i.e., label propagation and label spreading, where unlabeled data points are sampled or given. The superior results by our methods and elaborate analyses on the characteristics of DRE-BO-SSL exhibit the validity of our algorithms.

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

# A  ADDITIONAL COMPARISONS OF BORE AND LFBO

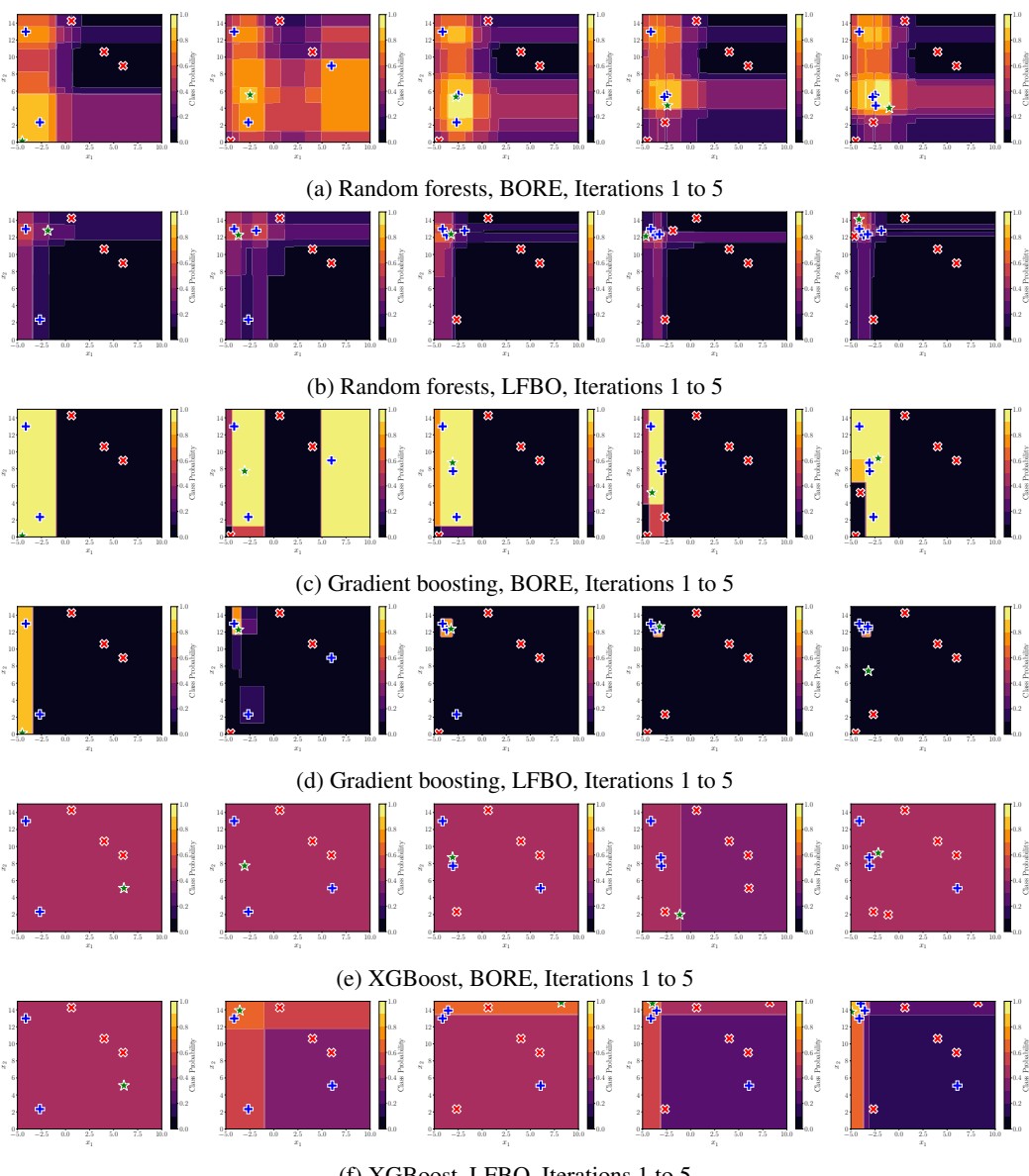

(a) Random forests, BORE, Iterations 1 to 5

(b) Random forests, LFBO, Iterations 1 to 5

(c) Gradient boosting, BORE, Iterations 1 to 5

(d) Gradient boosting, LFBO, Iterations 1 to 5

(e) XGBoost, BORE, Iterations 1 to 5

(f) XGBoost, LFBO, Iterations 1 to 5

Figure 8: Comparisons of BORE and LFBO by random forests, gradient boosting, and XGBoost for the Branin function. It follows the configurations described in Figure 1.

In addition to Figure 1, we include additional comparisons of BORE and LFBO by random forests (Breiman, 2001), gradient boosting (Friedman, 2001), and XGBoost (Chen & Guestrin, 2016) for the Branin function. For Figures 1 and 8, we use $\zeta = 0.33$ and $\beta = 0.5$.

# B  DETAILS OF DRE-BO-SSL

Algorithm 2 describes a procedure to label unlabeled data points; see the main article for the details of DRE-BO-SSL.

---

**Algorithm 2** Labeling Unlabeled Data

---

**Input:** Labeled data points $\mathbf{X}_l$, their labels $\mathbf{C}_l$, unlabeled data points $\mathbf{X}_u$, maximum iterations $\tau$, and a tolerance $\varepsilon$. Additionally, a clamping factor $\alpha$ for label spreading.

**Output:** Propagated labels $\widehat{\mathbf{C}}$.

  1: Initialize propagated labels $\widehat{\mathbf{C}}$ of $\mathbf{X}$.
  2: Compute similarities $\mathbf{W}$ and a degree matrix $\mathbf{D}$.
  3: Compute transition probabilities $\mathbf{P}$ with $\mathbf{W}$ and $\mathbf{D}$.
  4: **repeat**
  5:    Propagate $\widehat{\mathbf{C}}$ with $\mathbf{P}$ and the previous $\widehat{\mathbf{C}}$, and additionally $\alpha$ for label spreading.
  6:    Normalize $\widehat{\mathbf{C}}$ row-wise.
  7: **until** a change of $\widehat{\mathbf{C}}$ converging to $\varepsilon$ or reaching $\tau$.

---

## C   ANALYSIS OF DRE-BO-SSL

Under the cluster assumption, i.e., Assumption 4.1, a margin $\gamma$ is defined as a minimum distance between two decision boundaries.

**Definition C.1.** *Let a compact connected decision set be $\mathcal{C}_i \subseteq \mathcal{X}$ for $i \in \{0, 1\}$ and a boundary subset, i.e., a set of boundary points, of a compact connected set $\mathcal{S}$ be $\partial \mathcal{S}$. A margin $\gamma$ is defined as*

$$\gamma = (2\mathbb{I}(\mathcal{C}_1 \cap \mathcal{C}_2 = \emptyset) - 1) \min_{\mathbf{x}_1 \in \partial \mathcal{C}_1 \backslash \partial \mathcal{X}, \mathbf{x}_2 \in \partial \mathcal{C}_2 \backslash \partial \mathcal{X}} \|\mathbf{x}_1 - \mathbf{x}_2\|_2. \tag{11}$$

Using Definition C.1, we claim that a semi-supervised classifier in DRE-BO-SSL can mitigate the overconfidence problem presented in Section 1.1, because it can expand a decision set $\mathcal{C}_1$ for class 1 by reducing $\gamma$ with unlabeled data. However, we need to verify if a large decision set is derived from the characteristics of non-parametric classifiers, since a semi-supervised classifier we use is converted to the Nadaraya-Watson non-parametric model (Nadaraya, 1964; Watson, 1964) without unlabeled data. As shown in Figure 11, there is no strong relationship between the performance of semi-supervised classifiers without unlabeled data, i.e., the Nadaraya-Watson estimator, and one of semi-supervised classifiers with unlabeled data. It implies that configuration selection for a semi-supervised classifier is dependent on a class of objective function, and the presence of unlabeled data is likely to be effective for alleviating the overconfidence problem.

Singh et al. (2008) provide a sample error bound of supervised and semi-supervised learners, related to $\gamma$, $n_l$, $n_u$, and $d$. This work proves that a semi-supervised learner can be better than any supervised learners, by assuming that $n_u \gg n_l$ and access to perfect knowledge of decision sets. However, these theoretical results cannot be directly applied in our sequential problem because this work assumes that both labeled and unlabeled data points are independent and identically distributed. Nevertheless, these results can hint a theoretical guarantee on better performance of semi-supervised classifiers with unlabeled data points. Further analysis for the circumstances of Bayesian optimization is left for future work.

## D   EXPERIMENT DETAILS

Here we present the missing details of the experiments shown in the main part.

To carry out the experiments in our work, we use dozens of commercial Intel and AMD CPUs such as Intel Xeon Gold 6126 and AMD EPYC 7302. For the experiments on minimum multi-digit MNIST search, the NVIDIA GeForce RTX 3090 GPU is used.

To minimize (6) for finding an adequate $\beta$ of label propagation and label spreading, we use L-BFGS-B with a single initialization, which is implemented in SciPy (Virtanen et al., 2020), for all the experiments in Figures 2, 3, 5, 6, and 7. For the other empirical analyses, we set $\beta$ as a specific fixed value; see the corresponding sections for the details.

To reduce the computational complexity of DRE-BO-SSL for a scenario with a fixed-size pool, we randomly sample 2,000 unlabeled points from the predefined pool for all experiments excluding synthetic benchmark functions. More thorough analysis can be found in Section G.

To compare baseline methods with our methods, we assess optimization algorithms using a simple regret:

$$\text{simple regret}(f(\mathbf{x}_1), \ldots, f(\mathbf{x}_t), f(\mathbf{x}^*)) = \min_{i=1}^{t} f(\mathbf{x}_i) - f(\mathbf{x}^*), \tag{12}$$

where $\mathbf{x}^*$ is a global optimum.

## D.1 DETAILS OF BASELINE METHODS

As the competitors of our method, we test the following baseline methods:

- Gaussian process, EI and UCB: It is a Bayesian optimization strategy, which is defined with GP regression with the Matérn 5/2 kernel (Rasmussen & Williams, 2006). Expected improvement (Jones et al., 1998) and GP-UCB (Srinivas et al., 2010) are used as acquisition functions.
- Random forest, BORE and LFBO: These are BORE and LFBO that employ random forests (Breiman, 2001) with 1000 decision trees. Minimum samples to split are set to 2 for these baselines.
- Gradient boosting, BORE and LFBO: These methods are BORE and LFBO with gradient boosting classifiers (Friedman, 2001) with 100 decision trees. Learning rate for the classifier is set to 0.3.
- XGBoost, BORE and LFBO: Similar to gradient boosting, BORE and LFBO with XGBoost have 100 decision trees as base learners with learning rate of 0.3.
- MLP, BORE and LFBO: These methods are built with two-layer fully-connected networks. The detail of the multi-layer perceptron is described as follows.

The architecture of a multi-layer perceptron is

First layer: fully-connected, input dimensionality $d$, output dimensionality 32, ReLU;

Second layer: fully-connected, input dimensionality 32, output dimensionality 1, Logistic,

where $d$ is the dimensionality of the problem we solve.

Note that most configurations for the baseline methods follow the configurations described in the work (Song et al., 2022).

## D.2 DETAILS OF SYNTHETIC BENCHMARKS

Here we describe the details of synthetic benchmarks.

**Beale Function.** This function is defined as follows:

$$f(\mathbf{x}) = (1.5 - x_1 + x_1 x_2)^2 + (2.25 - x_1 + x_1 x_2^2)^2 + (2.625 - x_1 + x_1 x_2^3)^2, \tag{13}$$

where $\mathbf{x} = [x_1, x_2] \in [-4.5, 4.5]^2$.

**Branin Function.** It is defined as follows:

$$f(\mathbf{x}) = \left(x_2 - (5.1/4\pi^2)x_1^2 + (5/\pi)x_1 - 6\right)^2 + 10\left(1 - (1/8\pi)\right)\cos(x_1) + 10, \tag{14}$$

where $\mathbf{x} = [x_1, x_2] \in [[-5, 10], [0, 15]]$.

**Bukin6 Function.** This benchmark is given by the following:

$$f(\mathbf{x}) = 100\sqrt{|x_2 - 0.01x_1^2|} + 0.01|x_1 + 10|, \tag{15}$$

where $\mathbf{x} = [x_1, x_2] \in [[-15, -5], [-3, 3]]$.

**Six-Hump Camel Function.** It is given by the following:

$$f(\mathbf{x}) = \left(4 - 2.1x_1^2 + x_1^4/3\right)x_1^2 + x_1 x_2 + (-4 + 4x_2^2)x_2^2, \tag{16}$$

where $\mathbf{x} = [x_1, x_2] \in [[-3, 3], [-2, 2]]$.

Table 1: Search space for Tabular Benchmarks. "tanh" and "relu" represent the hyperbolic tangent and ReLU, respectively.

| Hyperparameter | Possible Values |
|---|---|
| The number of units for 1st layer | $\{16, 32, 64, 128, 256, 512\}$ |
| The number of units for 2nd layer | $\{16, 32, 64, 128, 256, 512\}$ |
| Dropout rate for 1st layer | $\{0.0, 0.3, 0.6\}$ |
| Dropout rate for 2nd layer | $\{0.0, 0.3, 0.6\}$ |
| Activation function for 1st layer | $\{$"tanh", "relu"$\}$ |
| Activation function for 2nd layer | $\{$"tanh", "relu"$\}$ |
| Initial learning rate | $\{5 \times 10^{-4}, 1 \times 10^{-3}, 5 \times 10^{-3}, 1 \times 10^{-2}, 5 \times 10^{-2}, 1 \times 10^{-1}\}$ |
| Learning rate scheduling | $\{$"cosine", "constant"$\}$ |
| Batch size | $\{8, 16, 32, 64\}$ |

### D.3 Details of Tabular Benchmarks

The search space of Tabular Benchmarks (Klein & Hutter, 2019) is described in Table 1. To handle categorical and discrete variables, we treat them as integer variables by following the previous literature (Garrido-Merchán & Hernández-Lobato, 2020).

### D.4 Details of NATS-Bench

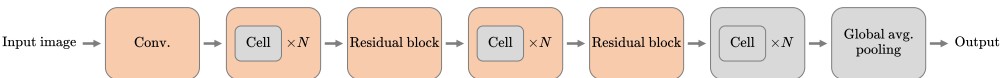

Figure 9: Neural network architecture in NATS-Bench. Orange blocks are optimized.

Table 2: Search space for NATS-Bench. There exist $8^5 = 32{,}768$ models.

| Hyperparameter | Possible Values |
|---|---|
| Output channels of 1st convolutional layer | $\{8, 16, 24, 32, 40, 48, 56, 64\}$ |
| Output channels of 1st cell stage | $\{8, 16, 24, 32, 40, 48, 56, 64\}$ |
| Output channels of 1st residual block | $\{8, 16, 24, 32, 40, 48, 56, 64\}$ |
| Output channels of 2nd cell stage | $\{8, 16, 24, 32, 40, 48, 56, 64\}$ |
| Output channels of 2nd residual block | $\{8, 16, 24, 32, 40, 48, 56, 64\}$ |

We describe the search space for NATS-Bench (Dong et al., 2021) in Figure 9 and Table 2.

### D.5 Details of Minimum Multi-Digit MNIST Search

Since multi-digit MNIST, which is composed of images of size $(28, 84)$, is high-dimensional, some of the methods used in this work, e.g., methods with random forests, gradient boosting, and XG-Boost, struggle to process such data. Therefore, we embed an original image to a lower-dimensional vector using an auxiliary convolutional neural network. The convolutional neural network is trained to classify a three-digit image to one of labels from "000" to "999," with the following architecture:

> First layer: convolutional, input channel $1$, output channel $8$, kernel size $3 \times 3$, padding $1$, ReLU, max-pooling $2 \times 2$;

> Second layer: convolutional, input channel $8$, output channel $16$, kernel size $3 \times 3$, padding $1$, ReLU, max-pooling $2 \times 2$;

> Third layer: convolutional, input channel $16$, output channel $32$, kernel size $3 \times 3$, padding $1$, ReLU, max-pooling $2 \times 2$;

> Fourth layer: fully-connected, input dimensionality $960$, output dimensionality $128$, ReLU;

> Fifth layer: fully-connected, input dimensionality $128$, output dimensionality $64$, ReLU;

Sixth layer: fully-connected, input dimensionality $64$, output dimensionality $1000$, Softmax.

The Adam optimizer (Kingma & Ba, 2015) with learning rate $1 \times 10^{-3}$ is used to train the network for 100 epochs. To train and test the model fairly, we create a training dataset of 440,000 three-digit images, a validation dataset of 40,000 three-digit images, and a test dataset of 80,000 three-digit images using a training dataset of 55,000 single-digit images, a validation dataset of 5,000 single-digit images, and a test dataset of 10,000 single-digit images in the original MNIST dataset (LeCun et al., 1998). For example, supposing that a test dataset has 1,000 single-digit images per class – it is not true for the MNIST dataset, but it is assumed for explanation – and we would like to generate a three-digit image "753," $10^9$ combinations for "753" can be created. We therefore randomly sample a fixed number of three-digit images from a vast number of possible combinations. In addition, an early stopping technique is utilized by comparing the current validation loss to the average of validation losses for the recent five epochs. Eventually, our network achieves $99.6\%$ in the training dataset, $97.0\%$ in the validation dataset, and $96.9\%$ in the test dataset.

To construct a fixed-size pool, we use 80,000 embeddings of dimensionality $64$, which are derived from the outputs of the fifth layer without ReLU, by passing the test dataset of three-digit images through the network.

## E  DISCUSSION ON A FREE PARAMETER IN LABEL PROPAGATION AND LABEL SPREADING

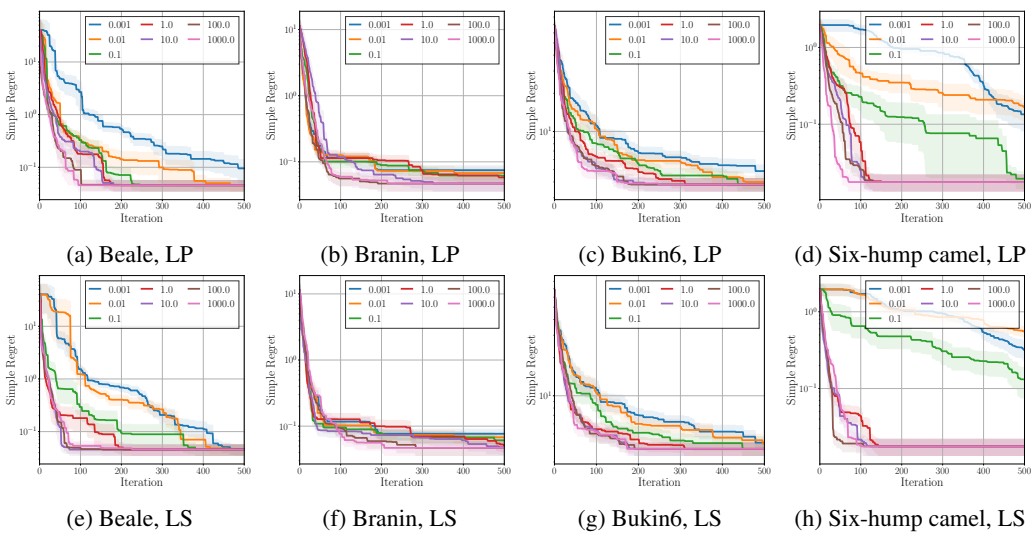

Figure 10: Effects of a free parameter $\beta$ in label propagation, denoted as LP, and label spreading, denoted as LS. All experiments are repeated 20 times.

In the Bayesian optimization process of DRE-BO-SSL, a free parameter $\beta$ in label propagation and label spreading is learned every iteration by minimizing (6); see Figure 4 for the results on learned $\beta$. Furthermore, to show the effects of $\beta$, we empirically analyze $\beta$ as depicted in Figure 10. We sample 1,000 unlabeled points and use all of them as unlabeled points without pool sampling. For the cases of four benchmark functions, higher $\beta$ tends to show better performance than lower $\beta$. These results considerably correspond with the results in Figure 4.

## F  DISCUSSION ON UNLABELED POINT SAMPLING

We design two studies to analyze the effects of the number of unlabeled points $n_u$ and sampling strategies in a process of unlabeled point sampling, where unlabeled points are not provided and $\beta = 0.5$ is given.

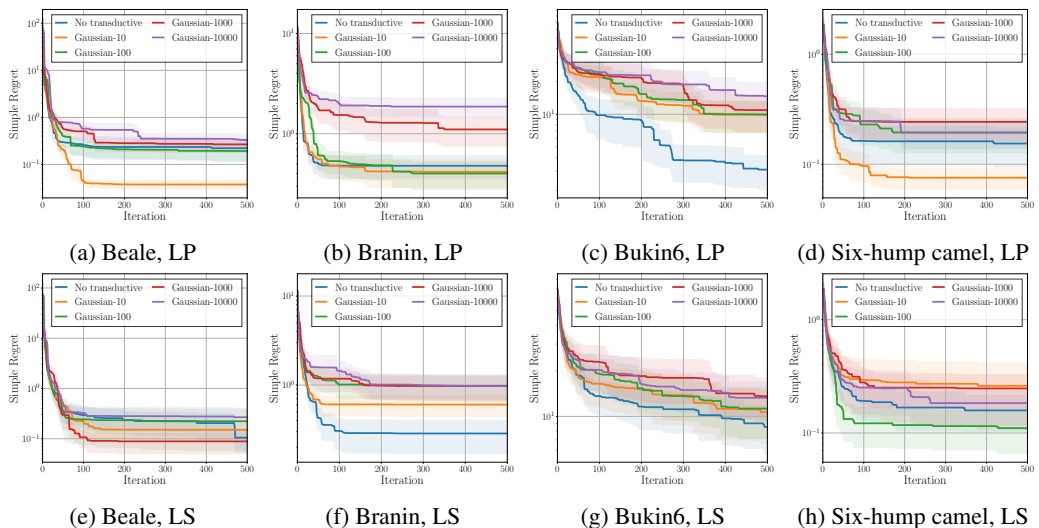

Figure 11: Effects of the number of unlabeled points for unlabeled point sampling. LP and LS stand for label propagation and label spreading, respectively. We repeat all experiments 20 times.

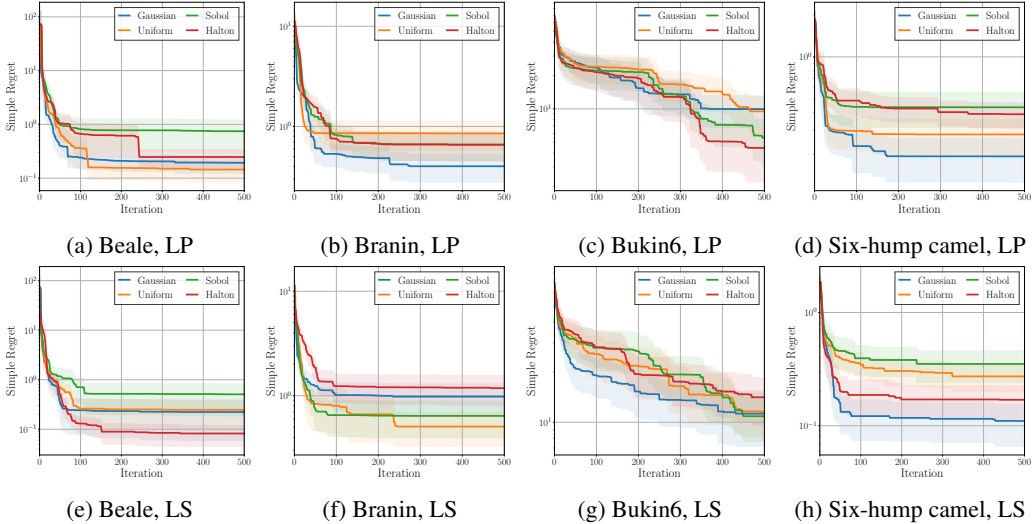

Figure 12: Effects of sampling strategies for unlabeled point sampling. LP and LS stand for label propagation and label spreading, respectively. We repeat all experiments 20 times.

For the first study, we conduct five settings, no unlabeled data, which implies that transductive learning is not used, and $n_u = 10, 100, 1000, 10000$. Interestingly, the tendency of the number of unlabeled points are unclear as presented in Figure 11. It implies that a setting for the number of unlabeled data points depend on the characteristics of benchmark functions, which is common in Bayesian optimization and black-box optimization. Besides, $\gamma$ is different across benchmarks and iterations and it lets optimization results sensitive to $n_u$. Therefore, we cannot determine a suitable setting without access to a black-box function of interest.

As another elaborate study, we test the effects of sampling strategies. Four strategies, the truncated multivariate normal distributions, uniform distributions, Halton sequences (Halton, 1960), and Sobol' sequences (Sobol', 1967), are compared. As depicted in Figure 12, the normal distribution is better than the other sampling methods in four cases and shows robust performance in most of the cases, but it is not always the best. Similar to the previous study on the effects of $n_u$, we presume that it is also affected by $\gamma$, which is hard to define in practice.

# G  DISCUSSION ON POOL SAMPLING

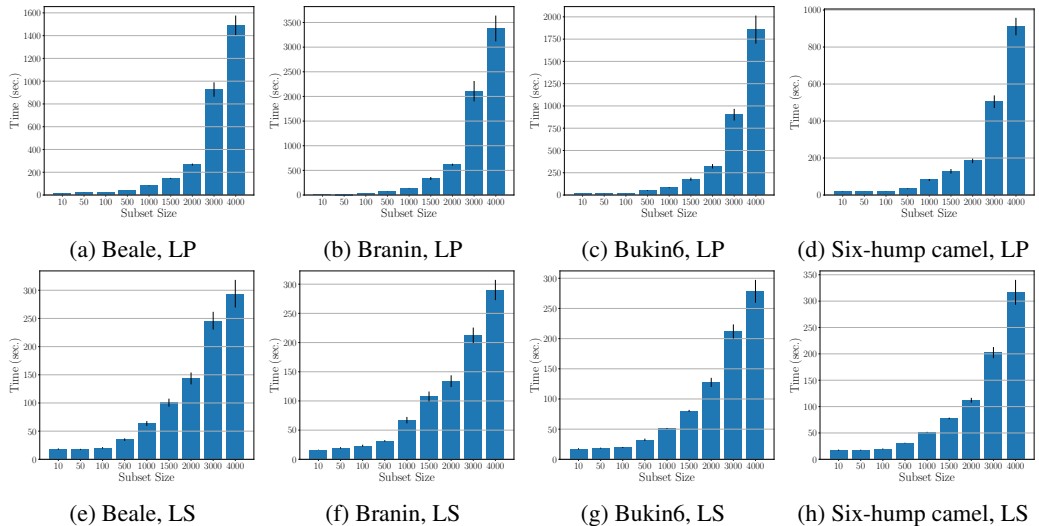

Figure 13: Results with 20 repeated experiments on elapsed times varying subset sizes via pool sampling. LP and LS stand for label propagation and label spreading, respectively.

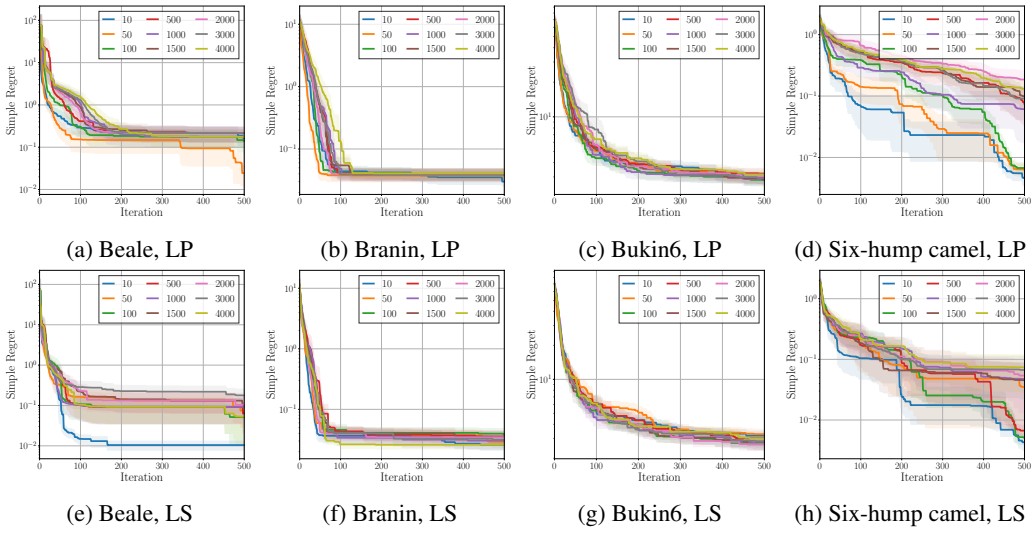

Figure 14: Effects of pool sampling for a case with fixed-size pools. We repeat all experiments 20 times, and LP and LS stand for label propagation and label spreading, respectively.

To see the impact of an additional hyperparameter, i.e., the size of a subset of the original pool, which is introduced to speed up semi-supervised learning algorithms, we demonstrate numerical analyses on pool sampling where the size of a predefined pool is 4,000 and $\beta = 0.5$ is given. Figure 13 reports elapsed times over subset sizes for pool sampling. Larger subsets make the framework slower as expected. Based on Figures 13 and 14, we can accelerate our framework without significant performance loss.

# H  LIMITATIONS

As discussed above, our algorithms slow down if a pool size is significantly large. As presented in Figure 13, elapsed times are certainly dependent on subset sizes. To tackle this issue, we suggest

a method to randomly select a subset of the pool, but a more sophisticated subsampling method can be devised for our framework. In particular, we can leverage the impacts of the subset of the pool by utilizing the geometric information of unlabeled data points.

## I   BROADER IMPACTS

Our work does not have a direct negative societal impact because it proposes a new Bayesian optimization method to optimize a black-box function. However, in the sense of optimization, this line of research can be used in optimizing any objective functions including harmful and unethical optimization tasks.

