# OpenReview forum: "Density Ratio Estimation-based Bayesian Optimization with Semi-Supervised Learning"
_ICLR.cc/2024/Conference — Submitted to ICLR 2024_

### Official Review · Reviewer_tEiT · 2023-10-30

**Soundness:** 2 fair
**Presentation:** 1 poor
**Contribution:** 2 fair
**Rating:** 6
**Confidence:** 4

**Summary:**

This paper proposes to use semi-supervised learning methods for Bayesian optimization (BO). The idea is to use an alternative paradigm for BO instead of fitting a regressor to the observed data. The authors suggest to use density ratio estimation BO instead. In DRE-BO one uses a classifier as the model to guide the search and the acquisition function is computed in terms of the class probability ratios. The authors suggest to strengthen the classifier accuracy by using semi-supervised learning techniques. The method is compared to other strategies in synthetic and real problems.

**Strengths:**

Extensive experimental evaluation.

**Weaknesses:**

The introduction is poor. It does not introduce properly the problem addressed in the paper.

In general the writing of the paper has to be improved a lot. It does not introduce the concept of DRE-based BO right. If the reader is not familiar with it, they cannot understand it properly. The authors have failed in this task.

It is not clear what is the motivation for DRE-based BO. It is also unclear how the threshold value y^t is chosen.

The authors have to better explain DRE-based BO, why does it work, why is it interesting and why it is better than regression based BO.

Figure 1 is not explained properly.

The proposed method is not very well motivated. It seems it is simply using a better classifier. The authors claim that they use semi-supervised learning techniques to train the classifier. However, the semi-supervised data seems to be generated by sampling from a truncated Gaussian and then label propagation is used to generate the associated labels. Therefore, the proposed method can be understood simply as using a better classifier. Given this, I do not find that much novelty in the proposed method.

**Questions:**

Please, explain better figure 1 and why it illustrates the over-confidence problem.

Please, explain how the threshold value y_t is chosen.

Please, explain why the proposed method needs to sample from a truncated multivariate Gaussian distribution.

---

> ### Author Response · Authors · 2023-11-20
> **Response to Reviewer tEiT (1/n)**
>
> We sincerely appreciate your constructive comment to improve our work.
>
> > The introduction is poor. It does not introduce properly the problem addressed in the paper.
>
> > In general the writing of the paper has to be improved a lot. It does not introduce the concept of DRE-based BO right.
>
> We will revise our paper more carefully in the final version.
>
> > It is not clear what is the motivation for DRE-based BO. It is also unclear how the threshold value $y^t$ is chosen.
>
> > Please, explain how the threshold value $y_t$ is chosen.
>
> We followed the previous work (Tiao et al., 2021; Song et al., 2022). In order to focus on the use of unlabeled points in the process of DRE-based Bayesian optimization, the selection of $y^t$ is built on the previous literature. Please refer to the prior work.
>
> > The authors have to better explain DRE-based BO, why does it work, why is it interesting and why it is better than regression based BO.
>
> We have included why DRE-based Bayesian optimization works in **Section 3**. Moreover, compared to GP-based Bayesian optimization, surrogate model learning is more efficient. In particular, while GP scales cubically with the number of points, learning a classifier is likely to be faster. Moreover, as shown in the experiment **Minimum Multi-Digit MNIST Search** of **Section 5.2**, the use of classifiers helps learn the representation of high-dimensional data.
>
> > Figure 1 is not explained properly.
>
> > Please, explain better figure 1 and why it illustrates the over-confidence problem.
>
> To accommodate your comment, we have updated **Section 1.1**. Please see **Section 1.1**.
>
> > The proposed method is not very well motivated. It seems it is simply using a better classifier. The authors claim that they use semi-supervised learning techniques to train the classifier.
>
> We do not agree with this comment. Our method offers a way to employ unlabeled points into the procedure of Bayesian optimization. In particular, in the scenario of pool-based Bayesian optimization, the vanilla Bayesian optimization does not utilize unlabeled points in a pool. Our method with semi-supervised learning enables us to utilize them in Bayesian optimization. Therefore, our contributions are more than simply using a better classifier.
>
> > Please, explain why the proposed method needs to sample from a truncated multivariate Gaussian distribution.
>
> The search space of Bayesian optimization is generally defined as a hypercube. Thus, we require utilizing a truncated multivariate Gaussian distribution so that the sampled points should be confined in the hypercube.

---

> > ### Comment · Reviewer_tEiT · 2023-11-22
> > **Response to authors**
> >
> > I acknowledge the efforts made by the authors for improving their paper and hence I have increased a bit my score in consequence.

---

> > > ### Author Response · Authors · 2023-11-22
> > >
> > > We appreciate your engagement in the discussion process.
> > >
> > > We will carefully revise the final version of our work based on your comment.

---

### Official Review · Reviewer_NoA7 · 2023-10-31

**Soundness:** 2 fair
**Presentation:** 1 poor
**Contribution:** 2 fair
**Rating:** 3
**Confidence:** 4

**Summary:**

This paper extends the binary classifier-based Bayesian optimization such that the classifier is trained in a semi-supervised manner. The authors argue that the semi-supervised classifier expands the region of ${\bf x}$ associated with high probability $P(y \leq y^\dagger | {\bf x}, \mathcal{D})$, which leads to more efficient exploration of black-box optimization.
Proposed method, called DRE-BO-SSL, is compared with the ordinary BO as well as DRE-BO with sueprvised classification approaches for function optimization as well as hyperparameter tuning tasks.

**Strengths:**

The idea sounds sensible to incorporate semi-supervised learning for encouraging the exploration of DRE-based Bayesian optimization method.
Figure 1 illustrates how the search space is explored by the proposed framework.
With that said, the figure needs more explanation for better comprehension of its content.
For example, the color bar is labeled as *Class Probability*, which suggests something like $p(y \leq y^\dagger | {\bf x}, \mathcal{D})$.
However, in the text most probability is shown as the distribution over the input such as $p({\bf x} | y \leq y^\dagger, \mathcal{D})$.
What does the figure specifically illustrate?

**Weaknesses:**

The presentation of current manuscript is problematic in that many things are uncertain from the text.
See *Questions* part below for details. The reviewer believes this information is necessary to better understand the method and enhance the reproducibility of results.

Unfortunately, the efficacy of proposed method in the performance of optimization over iterations is hardly distinguishable in Figs. 5 or 6.
I acknowledge a clear victory is not so common in comparisons of black-box optimization methods.
The bigger problem is that it is unclear  from the text on what condition and why the proposed method outperforms the existing approaches.

**Questions:**

1. Truncated normal distribution for sampling unlabeled data points.

How is matrix ${\bf A}$ designed? What covariance is taken for this disbiribution, for what?
What are lower and upper bounds ${\bf l}$ and ${\bf u}$?
Are they boundaries of search space $\mathcal{X}$? Is it assumed to be rectangular?

2. Fixed-size pool.

Does *fixed-size pool* scenario mean some finite number of candidate points ${\bf x}_i$ are selected in advance and that optimized over these points?
If true, is the set of points dynamically updated or fixed in advance?

3. Definition of simple regret.

What is the definition of simple regret?
Is it $f({\bf x}_n) - f^*$ in the $n$th iteration, where $f^*$ is the minimizer of function $f$.
I am wondering why the regret is monotonically decreasing in Fig. 2.
Does this plot $\min_n f({\bf x}_n) - f^*$?

4. What is dimensionality of ${\bf x}$ in four benchmark tasks in Fig. 2.

Are they all two? This is crucial information on the difficulty of black-box optimization problems.

5. Hyperparameter optimization of Fig. 6

How do you define the distance $\|{\bf x}_i - {\bf x}_j \|$, especially when categorical values are involved such as activation function and learning rate schedule.

---

> ### Author Response · Authors · 2023-11-20
> **Response to Reviewer NoA7 (1/n)**
>
> We sincerely appreciate your constructive comment to improve our work.
>
> > What does the figure specifically illustrate?
>
> We illustrated $p(\mathbf{x} | y \leq y^\dagger, \mathcal{D})$. We would like to emphasize that we do not model $y \leq y^\dagger$ and instead it indicates some class, e.g., Class 1, which implies points that might have function values smaller than $y^\dagger$. By following the conventional expression of machine learning, the probability we illustrated is expressed as $p(\mathbf{x} | c = 1, \mathcal{D})$ where the probability of interest is the probability of $c = 1$.
>
> > Unfortunately, the efficacy of proposed method in the performance of optimization over iterations is hardly distinguishable in Figs. 5 or 6.
>
> We disagree with this comment. Our methods quickly converge to solutions compared to other baseline methods. In particular, **Figure 6** shows the superior performance of our methods.
>
> > The bigger problem is that it is unclear from the text on what condition and why the proposed method outperforms the existing approaches.
>
> Based on the use of unlabeled points and semi-supervised learning, our methods can estimate more informative probabilities, thereby easing the over-confidence problem discussed in **Figures 1 and 8** and **Section 1.1**.
>
> > How is matrix $\mathbf{A}$ designed? What covariance is taken for this disbiribution, for what? What are lower and upper bounds $\mathbf{l}$ and $\mathbf{u}$? Are they boundaries of search space $\mathcal{X}$? Is it assumed to be rectangular?
>
> Covariance matrix was designed as an identity matrix, and then $\mathbf{A}$ can be calculated. Since the search space of Bayesian optimization is often assumed to be a hypercube, we also used a hypercube space. Thus, $\mathbf{l}$ and $\mathbf{u}$ can be determined by the search space given. We have updated our submission accordingly.
>
> > Does fixed-size pool scenario mean some finite number of candidate points $\mathbf{x}\_i$ are selected in advance and that optimized over these points? If true, is the set of points dynamically updated or fixed in advance?
>
> Yes, it is optimized over a fixed number of points. The set of points is not dynamically updated by following the general formulation of pool-based optimization.
>
> > What is the definition of simple regret? Is it $f(\mathbf{x}\_n) - f^*$ in the $n$th iteration, where $f^*$ is the minimizer of function $f$. I am wondering why the regret is monotonically decreasing in Fig. 2. Does this plot $\min\_{n} f(\mathbf{x}\_n) - f^*$?
>
> The definition of simple regret is $\min\_{n} f(\mathbf{x}\_n) - f^*$, which is widely used in comparing Bayesian optimization algorithms. We have included this definition; please see **Equation (12)**.
>
> > Are they all two? This is crucial information on the difficulty of black-box optimization problems.
>
> Yes, they are two-dimensional problems. We have added the specifics of the synthetic benchmarks; see **Section D.2**.
>
> > How do you define the distance $|\mathbf{x}\_i - \mathbf{x}\_j|$, especially when categorical values are involved such as activation function and learning rate schedule.
>
> To handle categorical and discrete variables, we treat them as integer variables by following the previous literature (Garrido-Merchan & Hernandez-Lobato, 2020). After converting them to integer variables, the distance between two integer variables is easily defined. We have updated **Section D.3** accordingly.

---

> > ### Author Response · Authors · 2023-11-22
> >
> > We deeply thank you for your valuable feedback.
> >
> > Please let us know if you have any concerns.

---

### Official Review · Reviewer_fMcY · 2023-11-01

**Soundness:** 2 fair
**Presentation:** 3 good
**Contribution:** 2 fair
**Rating:** 6
**Confidence:** 3

**Summary:**

The paper proposes a novel method called DRE-BO-SSL which combines SSL with DRE-based BO.
The intention is to improve the exploration-exploitation trade-off as previous DRE-based BO (BORE and LFBO) tends to focus on exploitation due to the over-confident classifiers.
The paper explores two types of SSL method (label propagation and label spreading).
Empirical results show that the proposed method work better than competitive BO methods on a wide range of tasks.

**Strengths:**

**originality** The proposed method is novel.

**quality** The proposed method is sound and the empirical results are promising.

**clarity** The technical part of the paper is good.

**significance** The proposed method is a good contribution to DRE-based BO and can be potentially useful to solve the over-confidence problem in DRE in general.

**Weaknesses:**

The presentation could be improved.
Specifically, the focus on over-confidence in the beginning is confusing and I only understand the main point until I read section 3.1 where the relation to exploitation is mentioned.
Perhaps this should be moved towards the front.

Some limitations of the work should be explicitly mentioned/discussed.
For example, assumption 4.1. seems to imply that the method is only intended to work on smooth functions.

Some related work is missing; see questions below.

**Questions:**

The over-confident problem of DRE is recently studied by [1,2].
Can the author(s) comment on how the construction of auxiliary distribution in these work is related to the sampling distribution of DRE-BO-SSL?

---

> ### Author Response · Authors · 2023-11-20
> **Response to Reviewer fMcY (1/n)**
>
> We sincerely appreciate your constructive comment to improve our work.
>
> > The presentation could be improved. Specifically, the focus on over-confidence in the beginning is confusing and I only understand the main point until I read section 3.1 where the relation to exploitation is mentioned. Perhaps this should be moved towards the front.
>
> Thank you for pointing this out. We moved Section 3.1 towards the front.
>
> > Some limitations of the work should be explicitly mentioned/discussed. For example, assumption 4.1. seems to imply that the method is only intended to work on smooth functions.
>
> We have described the limitations and societal impacts in **Sections H and I**. Moreover, we think that the limitation on the smoothness of an objective, which is mentioned by the reviewer, is also related to kernel selection. In Bayesian optimization common stationary kernels work well when an objective is smooth. Therefore, Assumption 4.1 does not significantly add the additional limitation of Bayesian optimization.
>
> > The over-confident problem of DRE is recently studied by [1,2]. Can the author(s) comment on how the construction of auxiliary distribution in these work is related to the sampling distribution of DRE-BO-SSL?
>
> We cannot find [1, 2] in your comment. Please let us know which papers you mentioned. We can add a response to this concern later.

---

> > ### Author Response · Authors · 2023-11-22
> >
> > We deeply thank you for your valuable feedback.
> >
> > Please let us know if you have any concerns.

---

> > ### Comment · Reviewer_fMcY · 2023-11-22
> >
> > Thanks for your response and revised draft.
> >
> > Here are the missing references I mentioned regarding the use of auxiliary distributions:
> > - [1] Telescoping Density-Ratio Estimation
> > - [2] Estimating the Density Ratio between Distributions with High Discrepancy using Multinomial Logistic Regression
> >
> > In particular, I was interested in seeing some relations between the sampling distribution of DRE-BO-SSL and their use of auxiliary in the sense that with the proposed sampling distribution the DRE problem "because harder" therefore avoids over-confidence (and performs better).

---

> > > ### Author Response · Authors · 2023-11-22
> > >
> > > Thank you for recommending interesting papers on density-ratio estimation.
> > >
> > > We think that the auxiliary distributions studied in [1,2] are closely related to our sampling distribution, which can guide our algorithm to a global solution more quickly. It might be analogous to the mechanism of mitigating distribution-shift problems in the papers you mentioned.
> > >
> > > We will add a discussion on this literature in the final version.

---

### Official Review · Reviewer_WaiC · 2023-11-01

**Soundness:** 3 good
**Presentation:** 3 good
**Contribution:** 3 good
**Rating:** 6
**Confidence:** 3

**Summary:**

This paper proposes an extension to bayesian optimisation (BO) using density rato estimation (DRE) to tackle the problem of overconfidence in the estimators used.  Specifically, the authors suggest using semi-supervised learning (transduction) to increase the accuracy of the model (overcome the difficulties typically caused by the imbalance between dataset sizes above and below the threshold).

**Strengths:**

The paper is very well written.  I quite like the underlying idea, and the implementation appears reasonable.

**Weaknesses:**

One doubt I have with this paper perhaps stems from unfamiliarity with semi-supervised algorithms in general.  My understanding of such approaches is that they tend to assume that the unlabelled training points are nevertheless generated from the underlying x distribution.  In most applications of BO, however, this concept is nonsensical: the only x distribution is the points sampled by BO, which are (in some sense) arbitrary (depending on the acquisition function they may cluster around the optimum as time passes, but not necessarily).  Am I missing a key point here?

**Questions:**

See weakness section.

---

> ### Author Response · Authors · 2023-11-20
> **Response to Reviewer WaiC (1/n)**
>
> We sincerely appreciate your constructive comment to improve our work.
>
> > One doubt I have with this paper perhaps stems from unfamiliarity with semi-supervised algorithms in general. My understanding of such approaches is that they tend to assume that the unlabelled training points are nevertheless generated from the underlying x distribution. In most applications of BO, however, this concept is nonsensical: the only x distribution is the points sampled by BO, which are (in some sense) arbitrary (depending on the acquisition function they may cluster around the optimum as time passes, but not necessarily). Am I missing a key point here?
>
> We have solved two scenarios, pool-based Bayesian optimization and generic Bayesian optimization (defined on a continuous search space). Notably, the formulation of pool-based Bayesian optimization has access to a fixed-size pool, which can be used as unlabeled points. The availability of the fixed-size pool makes DRE-based Bayesian optimization naturally built on semi-supervised learning. On the other hand, as pointed out by your comment, for the scenario of generic Bayesian optimization, the generation of unlabeled points may be nonsensical. However, our sampling process based on the truncated Gaussian distribution allows us to focus on the proximity of the query points that have already evaluated. This is aligned with the nature of Bayesian optimization, i.e., the consideration of both exploitation and exploration.

---

> > ### Author Response · Authors · 2023-11-22
> >
> > We deeply thank you for your valuable feedback.
> >
> > Please let us know if you have any concerns.

---

### Author Response · Authors · 2023-11-20
**General Comment to All Reviewers**

We thank the reviewers for their constructive feedback.

In particular, *Reviewer WaiC* mentioned that ***The paper is very well written*** and ***I quite like the underlying idea, and the implementation appears reasonable.*** *Reviewer fMcY* commented that ***The proposed method is novel***, ***The proposed method is sound and the empirical results are promising***, ***The technical part of the paper is good***, and ***The proposed method is a good contribution to DRE-based BO and can be potentially useful to solve the over-confidence problem in DRE in general***. *Reviewer NoA7* mentioned that ***The idea sounds sensible to incorporate semi-supervised learning for encouraging the exploration of DRE-based Bayesian optimization method***. *Reviewer tEiT* commented that ***Extensive experimental evaluation***.

Based on the reviewers' comments and suggestions, we have made the following improvements to our paper:

* Moved the Overconfidence Problem section to Section 1.1 (Overconfidence Problem).
* Updated Section 4.2 (Unlabeled Point Sampling).
* Added the definition of simple regret.
* Added Section D.2 (Details of Synthetic Benchmarks).
* Updated Section D.3 (Details of Tabular Benchmarks).
* Fixed minor issues.

---

### Meta-Review · Area_Chair_52vG · 2023-12-09

**Metareview:**

The paper deals with the problem of optimizing black-box functions via density ratio-based Bayesian optimization (BORE). The general weakness of such approaches is that they tend to be overconfident and lead to poor exploration. The paper proposes a simple semi-supervised learning to deal with such a deficiency, mainly by generating “pseudo-labels” that are needed for BORE.

All but one of the reviewers have highlighted that the presentation and clarity of the paper needs improvement. The authors have had a first attempt but I believe, as pointed out by the reviewers, it is still not great. Another important point is that it is unclear from the text on what condition and why the proposed method outperforms the existing approaches. Indeed, one of the reviewers did raise the issue of semi-supervised learning potentially being non-sensical in a general BO framework (the non-pool setting). I believe the paper needs some more grounded (conceptual, theoretical and empirical) justification for this. With this, I recommend (weak) rejection.

**Justification For Why Not Higher Score:**

Big issues with clarity and justification of the proposed methodology.

**Justification For Why Not Lower Score:**

N/A

---

### Decision · Program_Chairs · 2024-01-16

Reject